# Serum Scoring and Quantitative Magnetic Resonance Imaging in Intestinal Failure-Associated Liver Disease: A Feasibility Study

**DOI:** 10.3390/nu12072151

**Published:** 2020-07-19

**Authors:** Konstantinos C. Fragkos, María Claudia Picasso Bouroncle, Shankar Kumar, Lucy Caselton, Alex Menys, Alan Bainbridge, Stuart A. Taylor, Francisco Torrealdea, Tomoko Kumagai, Simona Di Caro, Farooq Rahman, Jane Macnaughtan, Manil D. Chouhan, Shameer Mehta

**Affiliations:** 1Intestinal Failure Service, Gastrointestinal Services, University College London Hospitals NHS Foundation Trust, London NW1 2BU, UK; konstantinos.fragkos@nhs.net (K.C.F.); mariaclaudiapicasso@gmail.com (M.C.P.B.); simona.dicaro@nhs.net (S.D.C.); farooq.rahman@nhs.net (F.R.); 2UCL Division of Medicine, University College London, London WC1E 6BT, UK; tomoko.kumagai.17@ucl.ac.uk; 3UCL Centre for Medical Imaging, University College London, London WC1E 6BT, UK; shankar.kumar@nhs.net (S.K.); lucy.caselton@nhs.net (L.C.); alex.menys.09@ucl.ac.uk (A.M.); stuart.taylor@ucl.ac.uk (S.A.T.); 4Department of Medical Physics, University College London Hospitals NHS Foundation Trust, London WC1N 3BG, UK; alan.bainbridge1@nhs.net (A.B.); francisco.torrealdea@nhs.net (F.T.); 5UCL Institute for Liver and Digestive Health, University College London, London WC1E 6BT, UK; j.macnaughtan@ucl.ac.uk

**Keywords:** parenteral nutrition, liver disease, intestinal failure, intestinal failure associated liver disease, fibrosis scores, magnetic resonance imaging

## Abstract

(1) Background: Intestinal failure-associated liver disease (IFALD) in adults is characterized by steatosis with variable progression to fibrosis/cirrhosis. Reference standard liver biopsy is not feasible for all patients, but non-invasive serological and quantitative MRI markers for diagnosis/monitoring have not been previously validated. Here, we examine the potential of serum scores and feasibility of quantitative MRI used in non-IFALD liver diseases for the diagnosis of IFALD steatosis; (2) Methods: Clinical and biochemical parameters were used to calculate serum scores in patients on home parenteral nutrition (HPN) with/without IFALD steatosis. A sub-group underwent multiparameter quantitative MRI measurements of liver fat fraction, iron content, tissue T1, liver blood flow and small bowel motility; (3) Results: Compared to non-IFALD (*n* = 12), patients with IFALD steatosis (*n* = 8) demonstrated serum score elevations in Enhanced Liver Fibrosis (*p* = 0.032), Aspartate transaminase-to-Platelet Ratio Index (*p* < 0.001), Fibrosis-4 Index (*p* = 0.010), Forns Index (*p* = 0.001), Gamma-glutamyl transferase-to-Platelet Ratio Index (*p* = 0.002) and Fibrosis Index (*p* = 0.001). Quantitative MRI scanning was feasible in all 10 sub-group patients. Median liver fat fraction was higher in IFALD steatosis patients (10.9% vs 2.1%, *p* = 0.032); other parameter differences were non-significant; (4) Conclusion: Serum scores used for non-IFALD liver diseases may be useful in IFALD steatosis. Multiparameter MRI is feasible in patients on HPN.

## 1. Introduction

Intestinal failure associated liver disease (IFALD) refers to a clinical state of hepatic injury arising as a result of intestinal failure (IF) and / or its treatment [1,2,3]. As a major complication of IF, it remains a major indication for intestinal transplantation (with or without liver transplantation) [1] and accounts for almost 30% of all deaths related to home parenteral nutrition (HPN) [4]. The aetiology is multifactorial where both parenteral nutrition (PN)- [5,6,7,8,9,10,11,12,13,14] and non-PN-related factors [10,15,16,17,18,19,20,21,22,23,24] contribute [25,26].

Recently, ESPEN defined IFALD as a complication occurring as a result of one or more factors relating to IF including PN and arising in the absence of another primary parenchymal liver pathology such as hepatotoxicity or biliary obstruction [3]. Once modifiable causes have been minimised, the definitive treatment for IFALD is intestinal transplantation. Isolated intestinal transplantation may reverse hepatic fibrosis [27] and UK experience indicates survival with isolated small bowel grafts is longer than that for liver-containing grafts [28,29]. Interestingly, the pathophysiology of IFALD in adults differs from the paediatric population, where steatosis with variable progression to fibrosis dominate, as opposed to predominantly cholestasis in children [25,30].

Despite the clinical importance, specific diagnostic criteria for IFALD in adults do not exist. Although liver biopsy is considered the gold standard, it is invasive with an attendant risk of complications, and therefore unsuitable for longitudinal monitoring. Furthermore, it is subject to sampling error and no validated consensus reporting tool for IFALD histology has been developed.

There is therefore a clinical need for accurate non-invasive markers both for diagnosis and disease progression from steatosis to fibrosis. Standard liver function tests (LFTs) are used routinely for monitoring patients with chronic IF, but deranged LFTs are have low specificity in IFALD, with reported prevalence in HPN populations of 5–85% [4,14,29,31]. Nine different diagnostic serum scoring systems for the presence of IFALD have been described, relating to different pathological subtypes [2,32], but none have been validated and their use has led to heterogeneity in the literature. Indeed, Sasdelli et al. [2] recently demonstrated wide variance in IFALD prevalence in a single HPN cohort depending on the diagnostic criteria used.

Composite serum scores have also been studied in relation to diagnostic and prognostic use in non-IFALD chronic liver diseases such as non-alcoholic fatty liver disease (NAFLD) and chronic viral hepatitides. The aspartate transaminase (AST)-to-platelet ratio index (APRI) carries moderate accuracy in predicting fibrosis and cirrhosis in patients with chronic hepatitis C [33,34], chronic hepatitis B [35,36,37] and NAFLD [38,39]. Data on the use of APRI in paediatric IFALD are conflicting, although it has been correlated with fibrosis [40,41,42,43]. In adult IFALD, APRI was correlated with biochemical and histologic cholestasis, but not fibrosis [44]. The Fibrosis 4 (Fib4) score correlates with fibrosis in chronic hepatitis B [45,46], NAFLD [47] and HIV/HCV co-infection [48], but in IFALD has been correlated with cholestasis rather than fibrosis [44]. Neither score has been specifically studied in relation to IFALD steatosis. The clinical utility of a number of other scores have been studied in various non-IFALD chronic liver diseases including the Forns index [49,50,51], Enhanced Liver Fibrosis (ELF) score [52,53,54,55,56], Fibrosis Index [49], gamma-glutamyl transferase (GGT)-to-platelet ratio (GPR) score [57] and the NAFLD fibrosis score [58,59,60], all of which have potential for clinical use but are not part of routine clinical practice. Finally, a few proprietary serum panels have also been assessed. Fibrotest has been shown to accurately predict fibrosis across a range of chronic liver diseases [61,62], while FibroMeter [63] and Hepascore [64] have been correlated with fibrosis in NAFLD and chronic Hepatitis C, respectively. The role of these scores in IFALD requires further investigation—it is possible that serum scores may be useful for the diagnosis/monitoring of IFALD, particularly as steatosis and fibrosis are pathophysiological phenomena common with other chronic liver diseases. It seems appropriate to assess the performance of these scores in the diagnosis of IFALD steatosis in the first instance, before assessing their use in disease progression to IFALD fibrosis.

Radiological assessment is routinely used in other chronic liver diseases including NAFLD and viral hepatitis. Ultrasound is readily available, inexpensive and accessible at the patient bedside, but quantitative approaches suffer from high inter- and intra-observer variability [65,66]. Results in IFALD patients are mixed—the detection of steatosis matches the results in the general population [2,67], but imaging-based quantitative measures have been unsuccessful in staging fibrosis [68,69]. Biomechanical sonographic methods such as transient elastography, in widespread use for the assessment of liver fibrosis in non-IFALD chronic liver diseases, have in adult IFALD patients correlated with histological cholestasis and serum bilirubin but not fibrosis [44]. Acoustic radiation force impulse ultrasound elastography has been more promising, with a biopsy-validated differentiation between absent/mild and moderate/severe fibrosis in paediatric IFALD patients [70].

Magnetic resonance imaging (MRI) has an established role in the assessment of chronic liver diseases, but the use of quantitative MRI methods for the assessment of chronic liver disease remains largely in the research rather than clinical setting. Proton magnetic resonance spectroscopy (^1^H-MRS) can be used for reference standard measurements of hepatic steatosis [68,71] and has demonstrated increased fat–water ratios in patients with IFALD [28,29]. Chemical shift imaging-based measurements of the liver fat fraction have reported a prevalence of steatosis of 28.6% in patients with chronic IF, but it is unclear how many of these patients had IFALD [72]. In any case, ^1^H-MRS measurements of liver fat have in current practice been superseded by proton density fat fraction (PDFF) measurements [73] and, while a study indirectly inferring these measurements in IFALD patients have estimated moderate–severe steatosis (average 19.6% liver fat fraction) [74], no formal liver MRI PDFF measurements data in patients with IFALD have been reported.

Once patients are in the scanner, MRI also provides the opportunity to explore the use of other quantitative MRI methods for the evaluation of chronic liver disease. For example, liver PDFF measurements rely on T2* mapping data which can also be used to derive robust estimates of liver iron concentration [73,75], and may be of interest in assessing metabolic consequences of long-term parenteral nutrition [76]. Measurements of liver T1 have been correlated with fibrosis in chronic liver disease [77] and measurements of portal venous and hepatic arterial blood flow using phase-contrast MRI have been correlated with the severity of portal hypertension [78]. In conjunction, these measures have the potential to yield pathophysiological insights into IFALD.

Imaging of the abdomen can also enable the severity of MRI-quantified liver disease to be evaluated with other potential markers of intestinal tract dysfunction such as small bowel motility. The quantification of small bowel motility using cine MRI has proven useful in the assessment of inflammatory bowel disease [79], and while dysmotility is a reported feature of IF [80], there are no published MRI small bowel motility data in patients with IFALD.

Finally, probing multiple organs with different quantitative MRI methods enables the assimilation of these measurements to derive composite scores—so-called ‘multiparametric MRI’— with the potential to yield more robust and comprehensive indices of disease than individual quantitative MRI measurements in isolation.

There is therefore a clear need to develop robust non-invasive markers for routine clinical use in IFALD. The aims of this study were (i) to assess the clinical utility of serum scores used in the management of non-IFALD chronic liver diseases for the diagnosis of IFALD steatosis, and (ii) to assess the feasibility of performing multiparameter quantitative MRI studies in a cohort of patients on HPN and explore differences in these parameters in IFALD steatosis patients.

## 2. Methods

### 2.1. Study Design and Participants

This was a single-centre, cross-sectional study of patients with type 3 IF on HPN. Patients were identified from the IF database at University College London Hospitals (UCLH). Electronic medical records were interrogated, including results of previous radiological tests. Data on patients’ demographics, history of disease, clinical treatment, medications, and nutritional therapy were collected. To determine potential cofounders, clinical and nutritional variables were recorded. Clinical variables included pathophysiological classification of IF, small bowel length, co-morbidities, presence of remaining colon, use of hepatotoxic medications, use of antibiotics, prevalence of catheter related blood stream infection episodes, use of Taurolock ^TM^ and Curos ^TM^ disinfecting port protectors and results of hydrogen breath tests assessing for small intestinal bacterial overgrowth. The following nutritional variables were measured: age of initiation of PN, PN duration, number of days of PN infusion per week, number of days of PN lipid infusion per week, dose of PN lipid infusion, type of PN lipid preparation, mean kilocalories in PN solutions per day, presence of oral intake and presence of enteral intake. Patients were classified into an ‘IFALD steatosis’ group if ultrasound (US), computed tomography (CT), MRI or magnetic resonance cholangiopancreatography (MRCP) imaging performed within the previous six months had reported parenchymal liver steatosis in the absence of cholestasis or advanced liver disease (inclusive of a second review) [1]. Patients were classified into a ‘non-IFALD’ group if imaging performed within the previous six months had reported no parenchymal liver abnormalities. Exclusion criteria were malignancy, lack of hepatic imaging in the preceding six months or imaging demonstrating cholestasis or advanced liver disease.

### 2.2. Laboratory Tests

For the serum study, blood samples were collected at UCLH during planned hospital visits. Serum levels of haemoglobin, white cell count, platelet count, sodium, potassium, urea, creatinine, estimated glomerular filtration rate, magnesium, phosphate, vitamin D, glucose, total cholesterol, triglycerides, and C-reactive protein were measured. In addition, liver function tests including bilirubin, AST, alanine transaminase (ALT), alkaline phosphatase (ALP), GGT, albumin, international normalized ration (INR), and prothrombin time were evaluated. For the purpose of the study, mean levels of bilirubin, AST, ALT, ALP, and GGT, from the last six months were used. Other serum tests associated with liver function such as antimitochondrial antibodies, total bile acids, procollagen III N-terminal peptide (PIIINP), haptoglobin, apolipoprotein A1 and ferritin were also measured. Composite serum scores were also measured at a single time point as shown in Table 1.

### 2.3. MRI Scanning

For the MRI feasibility study, patients from both IFALD and non-IFALD groups were approached. Demographic and clinical data were collected as per the serum study. Local ethics committee approval was obtained (UCL Research Ethics Committee, Approval no. 07/Q0502/15), and all participating patients provided informed written consent. Patients were eligible to participate in the MRI feasibility study if they had no MRI contraindications (such as claustrophobia or previous non-MR safe metallic implants). All scans were undertaken in the afternoon to allow time for patients to attend after morning PN discontinuation. Scans were scheduled on the same day as outpatient clinical appointments to maximise patient convenience.

Imaging was performed on a 3T scanner (Ingenia, Philips Healthcare, Best, Netherlands), with a 16-channel body coil (SENSE XL Torso, Philips Healthcare, Best, Netherlands). Briefly, a multiparametric MRI protocol was used that included (a) anatomical axial, coronal and sagittal breath-hold balanced steady-state free precession imaging for anatomical planning, (b) liver PDFF measurements images and T2* maps for measurement of liver iron concentration using the Philips mDixon Quant sequences, (c) liver T1 mapping using multi-inversion time coronal spectral pre-saturation with inversion recovery spin echo sequences [82], (d) caval subtraction two-dimensional phase-contrast MRI measurements of portal venous, infrahepatic and suprahepatic inferior vena caval flow [83] and (e) coronal dynamic cine balanced fast field echo imaging with coverage of the whole small bowel volume for measurement of small bowel motility [84]. Full methodological acquisition and post-processing details can be found in Appendix A. All MRI scanning was undertaken by a radiographer (L.C.) with over 5 years of experience in abdominal imaging and trained in planning caval subtraction phase-contrast MRI studies.

The primary outcome was the number of eligible consenting patients who completed full MRI protocols with all datapoints collected (target 80% with study failure if <50% patients underwent MRI scans or <50% datapoints were collected). Secondary exploratory analyses of measurements of liver PDFF measurements, liver iron concentration, and liver T1 were based on averages from regions of interest (ROIs) drawn on each of the nine Couinaud liver segments, ROIs drawn on each vessel for bulk flow phase-contrast MRI measurements and manual segmentation of small bowel for small bowel motility. Post-processing and segmentation processes are described in detail in Appendix A. ROI placement and segmentation was undertaken by a hepato-pancreatico-biliary radiologist (M.D.C.) with over 10 years of experience in abdominal imaging, blinded to the patient group.

### 2.4. Statistical Analysis

Data were analysed using SPSS software package version 24.0 (SPSS, Inc., Chicago, Illinois, USA) and R 4.0. Mean and standard deviation (SD) or median and range were used to summarize values. Shapiro–Wilks tests were performed to test normal distribution of numerical data. Comparison of mean/medians between groups were done using *t*-tests for parametric analyses, Mann–Whitney U-test for non-parametric analyses, and Wilcoxon signed rank for comparing against a hypothetical median. Frequency data were compared using Fisher’s exact test. *p*-values less than 0.05 were considered statistically significant. Sample size and power calculations were performed based on a study measuring APRI in paediatric patients with IF [40]. Calculation using an α-value of 0.05 and a β-value of 0.80 reported a minimum sample size of 22 subjects, 11 for each group, for the present study.

## 3. Results

### 3.1. Serum Cohort

#### 3.1.1. Patient Demographics

Twenty patients were divided between IFALD steatosis (*n* = 8) and non-IFALD (*n* = 12) groups as defined above. The clinical and nutritional characteristics of the two groups are shown in Table 2. Both groups were matched for age, pathophysiological classification, small bowel length, oral diet, PN energy, lipid use and number of infusions per week. Participants in the IFALD steatosis group had significantly higher BMIs (23.73 ± 3.36 kg/m^2^) than participants with no IFALD (19.91 ± 2.95 kg/m^2^). A trend towards longer PN duration was observed in the IFALD steatosis group, although this difference was not significant. In no patients of the IFALD steatosis group were there any radiological or biochemical evidence of cholestasis, fibrosis, cirrhosis or portal hypertension. Similarly, there was no evidence of liver disease (including NAFLD) prior to the development of intestinal failure in any patient.

#### 3.1.2. Biochemical Parameters

Mean values of serum markers over the preceding six months were calculated for each patient and reported as means for each group, with results shown in Table 3 and Figure 1. Serum AST, ALT, GGT and PIIINP were all significantly elevated and outside of the normal range in the IFALD steatosis group. Serum platelet count, bilirubin and haptoglobin were also significantly different between the two groups but with mean values within the normal range. The following range of other serum variables were not found to be significantly different between the two groups: estimated glomerular filtration rate, magnesium, phosphate, vitamin D, total cholesterol, sodium, potassium, urea, haemoglobin, creatinine, white cell count, glucose, triglycerides, bile acids, ALP, ferritin, albumin, INR, prothrombin time, and Apoliprotein A1.

#### 3.1.3. Composite Serum Scores

The most contemporaneous blood results were used to calculate composite serum scores, as shown in Table 1. Results are shown in Table 4 and Figure 1. The APRI score (*p* < 0.001), Fib4 score (*p* = 0.010), Forns Index (*p* = 0.001), GPR score (*p* = 0.002), ELF score (*p* = 0.032) and Fibrosis Index (*p* = 0.001) means were all significantly higher in the IFALD steatosis group than the non-IFALD group. No significant difference between IFALD steatosis and non-IFALD participants were reported when measuring the AST/ALT ratio and the NAFLD score.

### 3.2. MRI Cohort

#### 3.2.1. Demographics

Ten patients from the serum cohort were taken forward for the MRI study and grouped into IFALD steatosis (*n* = 5) and non-IFALD (*n* = 5) groups. Demographics and clinical characteristics are shown in Table 5.

#### 3.2.2. Feasibility Results

Results of the proportion of patients progressing through the feasibility study are shown in Figure 2. Of the 10 patients, nine were able to attend on pre-agreed dates and times for scanning. For one patient, the scan was delayed by 9 weeks. This was due to both an inability to attend on agreed dates and a late nursing HPN disconnection resulting in hospital transport being missed. All scans were performed on the same day as IF clinic appointments. No PN infusions were missed as a result of the scan. T1 mapping data were not of sufficient quality for one patient (due to scanner-related technical factors) and small bowel motility data were not obtained for two patients (due to scanning protocol errors). All other imaging data were of sufficient quality for quantification.

#### 3.2.3. Quantitative Parameters

Median liver fat fraction was significantly higher in IFALD steatosis patients (10.90%, range 2.2–27.4%, vs non-IFALD 2.14%, range 1.4–4.2%, *p* = 0.032, Figure 3), but increased median liver iron concentration in IFALD steatosis patients (16.0 μmol/g, range 11.3–46.7 μmol/g vs non-IFALD 11.3 μmol/g, range 10.4–38.9 μmol/g, *p* = 0.222) and median liver T1 in IFALD steatosis patients (740 ms, range 594–919 ms, vs non-IFALD 715 ms, range 544–848 ms, *p* = 0.873) were not statistically significant. Reduced median portal vein flow in IFALD steatosis patients (56.2 mL/min/100g, range 37.5–93.0 mL/min/100g, vs non-IFALD 64.2 mL/min/100g, range 45.6–137.6 mL/min/100g, *p* = 0.667), estimated total liver blood flow in IFALD steatosis patients (62.1 mL/min/100g, range 27.9–119.3 mL/min/100g, vs non-IFALD 91.8 mL/min/100g, range 63.5–145.7 mL/min/100g, *p* = 0.151) and small bowel motility in IFALD steatosis patients (0.16 a.u., range 0.11–0.21 a.u., vs non-IFALD 0.19 a.u., range 0.07–0.21 a.u., *p* > 0.999) were not statistically significant (Table 6). Figure 3, Figure 4, Figure 5 and Figure 6 demonstrate examples of T2* and T1 parametric mapping (Figure 4), caval subtraction phase-contrast MRI (Figure 5) and small bowel motility studies (Figure 6).

## 4. Discussion

IFALD remains an important complication of chronic IF, yet agreed diagnostic criteria are lacking. In the absence of specific pharmacological interventions, intestinal transplantation is the definitive treatment, although decision making regarding the timing of transplantation can be difficult. The principle indications for transplantation in this context are progressive hepatic fibrosis or cirrhosis, for which the gold standard investigation remains liver biopsy. This procedure is associated with risk and therefore there remains a need to develop accurate non-invasive markers to stratify patients at high risk of fibrosis to inform use of liver biopsy. There has been a rapid advancement in the development of such non-invasive markers of disease progression in non-IFALD chronic liver diseases. Serum markers have been particularly widely studied, with a range of scores assessed in various pathologically distinct chronic liver diseases (Table 1). However, before considering longitudinal studies of serum markers alongside disease progression in adult IFALD, it is important to assess the utility of these markers in the diagnosis of IFALD steatosis at baseline. We therefore sought to understand the value of studying these serum scores in a larger prospective cohort by performing a retrospective study in a small single centre cohort.

Patients in our study were characterised as IFALD steatosis, in the absence of features of fibrosis, cholestasis or cirrhosis. The serum scores in Table 4 were found to be significantly elevated in the IFALD steatosis group when compared to the matched non-IFALD chronic IF group. These scores are all non-propriety and therefore can be simply calculated using routinely available laboratory tests. This appears initially counterintuitive given that these scores correlate with fibrosis to varying degrees in non-IFALD chronic liver diseases. However, the primary intention of previous association studies was with fibrosis or advanced liver disease, rather than with well-phenotyped steatosis cohorts. Additionally, it is possible that subclinical fibrosis was present in some patients in the present study; no patients had undergone prior liver biopsy. Furthermore, the pathophysiology of IFALD is much less well characterised than other chronic liver diseases. Whilst it is possible that the development of IFALD relates more closely to that of NAFLD than viral hepatitis, this has not been formally established. For example, little is known about vascular dynamics, the pathology of iron deposition, the function of hepatic stellate cells and other processes in IFALD. It is possible that some, or a combination, of these factors explain the serum score correlations found both in this small study and other non-IFALD studies.

A large range of values was observed in both groups, which may be a function of the small sample size or that subclinical fibrosis was present in some patients. The mean PN duration for the IFALD steatosis group was just over seven years, with none of the patients having progressed to fibrosis. This may be due to proactive management by the treating IF team but may also suggest that these patients had biologically ‘non-progressive’ disease, perhaps representing a more benign group than would otherwise be expected in a larger cohort. Overall, these results do suggest a larger study is warranted, particularly recruiting patients at the first point of diagnosis of IFALD steatosis, but also those with a range of hepatic injury. Further studies assessing for metabolic derangements as observed in NAFLD (e.g., insulin resistance) may be complementary.

Imaging modalities such as ultrasound, standard protocol MRI and MRCP are central in the diagnosis and surveillance of various non-IFALD chronic liver diseases. In this study, we propose a multiparametric quantitative MRI protocol that includes hepatic PDFF, T2* and T1 mapping, with caval subtraction PCMRI and small bowel cine MRI as these aim to measure liver fat, iron content, fibrosis, blood flow and small bowel motility, all of which are relevant pathophysiological factors in IFALD. This study sought to understand whether it was possible to perform multiparametric MRI in patients with chronic IF, a group who are often burdened with logistical issues impacting on clinical care, and whether recognisable outputs relating to IFALD could be obtained. We found that all patients approached proceeded to consent for the study and successfully underwent scanning with a high proportion of quantitative MRI parameters obtained. This required a proactive approach from the clinical and radiological teams: arranging scans on the same day as other hospital visits to minimise disruption, prioritising afternoon appointments to allow sufficient travel time after morning HPN disconnections, etc. Patients found the investigation acceptable and did not experience any adverse events relating to PN administration or venous catheters.

MRI PDFF measurements confirmed expected significant steatosis in the IFALD steatosis group, with four out of five in the IFALD steatosis group meeting the diagnostic criteria for non-alcoholic fatty liver disease (>5% as per guidelines [71]), but no statistically significant differences were observed with other quantitative MRI parameters.

Liver iron concentration measurements confirmed no significant iron overload in either group, which may reflect active clinical management of patient nutritional requirements. Historically, long-term PN has been associated with iron overload [85] but, more recently, PN-related iron deficiencies [86] have also been reported in the literature. It is worth noting that infusions of ferric compounds (and secondarily high ferritin levels) have been shown to affect MRI signal in patients receiving HPN [67]. The relationship between IFALD and liver iron concentration remains unclear, but a mechanistic overlap between the onset of hepatic steatosis and siderosis has been reported [87] and may yet be relevant as understanding of IFALD increases.

Liver T1, a measure of liver fibrosis, was also expectably similar in both groups and within previously reported ranges for non-fibrotic liver [88], confirming that none of the patients had progressed to fibrosis.

The relationship between hepatic blood flow measurements and CLD are complex, not least because short-term fluctuations in portal venous flow are closely related to prandial state [89,90]. Hepatic blood flow measurements are therefore normally made in the fasted state, but in this study, logistic factors were prioritised over ensuring a 4-h interval after HPN disconnection. Measures such as portal vein flow may also be significantly affected by extra-hepatic factors such as small bowel length or PN duration, neither of which were controlled for between groups. Pre-clinical studies and studies in patients with chronic liver disease have demonstrated PN-related reductions in portal vein flow [91,92]. Our study demonstrated lower caval subtraction phase-contrast MRI measurements of portal vein and total liver blood flow in IFALD steatosis patients, but these differences were not statistically significant. A wide variance of portal vein flow measurements and propagated errors in the caval subtraction phase-contrast MRI measurements [83] make these measurements prone to sample size-related type I/II errors.

Bowel motility disorders are a cause of IF and have been associated with IFALD [93]. Although short bowel syndrome patients are also likely to have abnormal small bowel motility [80], no formal quantitative studies have been reported. Abnormal intestinal motility is known to predispose to bacterial overgrowth [94], which in turn has been associated with the severity of chronic liver disease [95]. Here, we did not demonstrate any difference in MR small bowel motility in patients with IFALD steatosis, indeed reported values in both patient groups were similar to those previously reported in healthy volunteers [84]. Whilst this may reflect the mild severity of liver disease in our MRI study or variations in other factors including medications, this finding (while negative) is itself of interest in the broader context of IF.

In this exploratory study, we have proposed a range of quantitative MRI techniques, but these are not exhaustive and other quantitative methods, such as intra-voxel incoherent motion, tissue perfusion measurement using dynamic contrast enhanced/arterial spin labelling and MR elastography may also have a place in the diagnosis and surveillance of IFALD.

Although multiparametric MRI has greater resource implications than ultrasound, it provides more comprehensive information that may be useful particularly as our understanding and diagnostic criteria for IFALD evolve. Most hospitals have access to MRI scanning and more complex post-processing requirements could be met by data sharing and centralised data analysis protocols. It may be worthwhile, therefore, given this general applicability, to formally investigate the role of this MRI in the diagnosis of IFALD steatosis and IFALD fibrosis in a prospective study. Indeed, it is possible that composite scoring using combinations of serum markers as presented here with MRI outputs may prove more valuable than either alone.

The limitations of this study include the retrospective design, small sample sizes and single centre cohort. The cohort sizes did not allow for matching of many variables. Hepatic transient elastography is not routinely performed for patients with IFALD at our hospital and so it was not possible to correlate with serum scores. Ideally, even feasibility studies would typically involve greater numbers of participants but the overall numbers of patients with IFALD in any one centre will not be sufficiently high to allow for this. Selection bias was minimised by not excluding patients without available liver histology which would potentially involve only the most severe cases.

## 5. Conclusions

In conclusion, non-invasive diagnosis and monitoring of IFALD is an important unmet clinical need. There has been rapid development of non-invasive monitoring in non-IFALD chronic liver diseases. Given the pathological overlap with IFALD, it would seem reasonable to extend the study of IFALD markers to this group. A multiparametric IFALD-specific quantitative MRI protocol is an attractive potential imaging modality in IFALD, and this study suggests it is feasible to perform in patients on HPN. Indeed, imaging outputs may be complementary to serum scoring. Validating the associations of serum markers used here and examining the utility of multiparametric MRI in a larger prospective study would be a valuable exercise. This should ideally be on a national, multicentre basis given the insufficient numbers of patients with distinct IFALD pathologies in any single unit for meaningful analysis.

## Figures and Tables

**Figure 1 nutrients-12-02151-f001:**
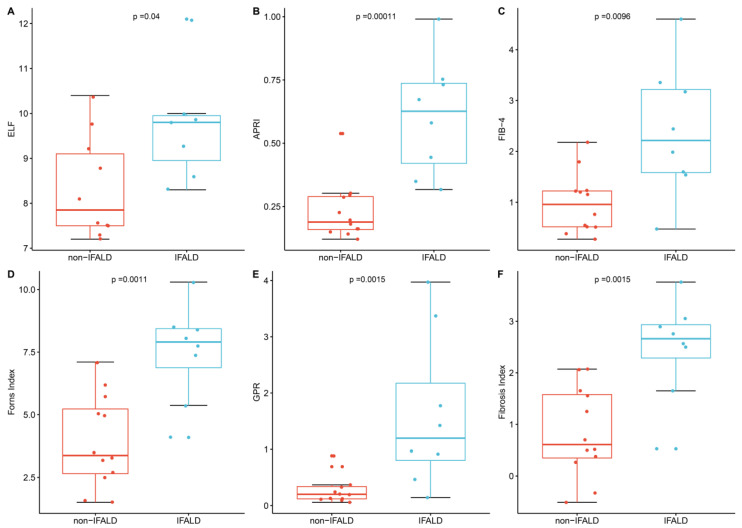
Comparison of liver scores between Intestinal Failure-Associated Liver Disease (IFALD) patients and non-IFALD patients, with respect to (**A**) Enhanced Liver Fibrosis (ELF); (**B**) Aspartate transaminase-to-Platelet Ratio Index (APRI); (**C**) Fibrosis 4 (FIB-4); (**D**) Forns Index; (**E**) Gamma-glutamyl transferase-to-Platelet Ratio (GPR); and (**F**) Fibrosis Index.

**Figure 2 nutrients-12-02151-f002:**
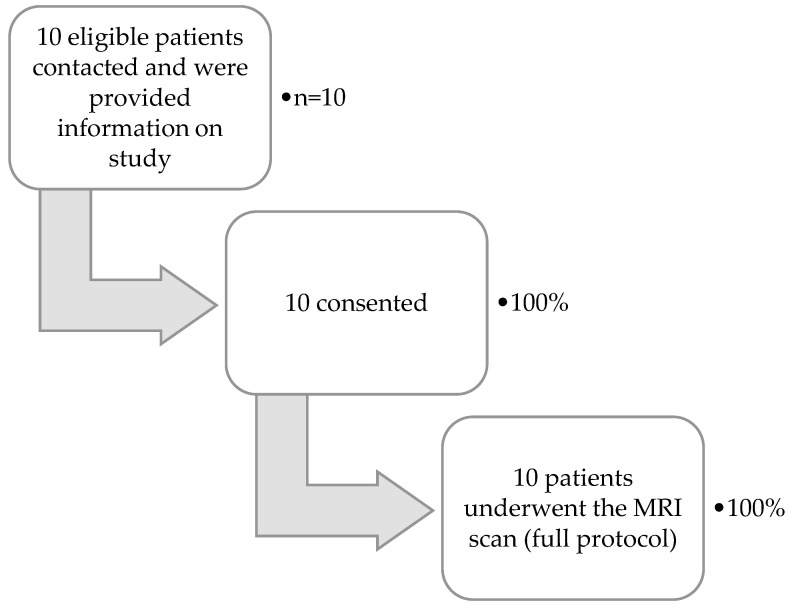
Magnetic resonance imaging (MRI) cohort flow chart.

**Figure 3 nutrients-12-02151-f003:**
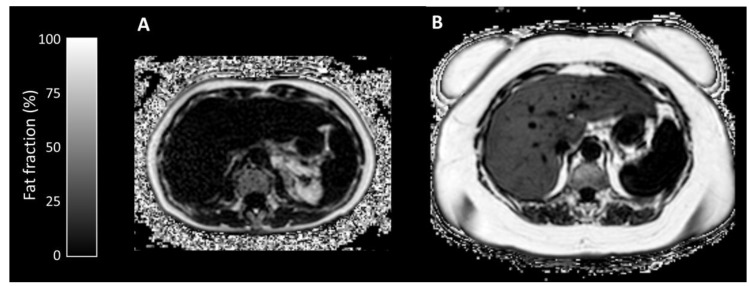
Axial MRI Proton Density Fat Fraction map examples from (**A**) non-IFALD and (**B**) IFALD steatosis patients. Average liver fat fraction was (**A**) 1.4% for the non-IFALD patient as represented by the darker liver parenchyma and (**B**) 27.4% for the IFALD steatosis patient as represented by the brighter liver parenchyma. Note the striking disparity in subcutaneous adiposity between patients.

**Figure 4 nutrients-12-02151-f004:**
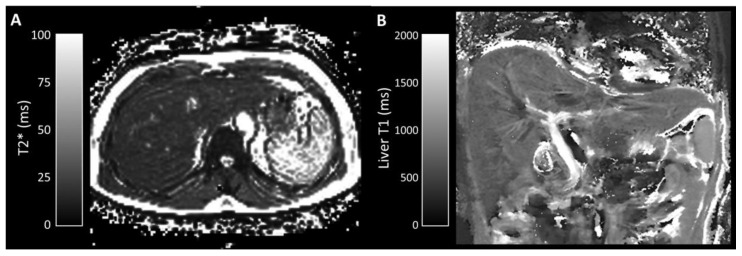
Parametric mapping of (**A**) liver T2* (axial) and (**B**) liver T1 (coronal). Segmental regions of interest placed (**A**) on T2* maps were converted into liver iron concentration (average 13.6 μmol/g in this example) and (**B**) on T1 maps (average 848 ms in this example).

**Figure 5 nutrients-12-02151-f005:**
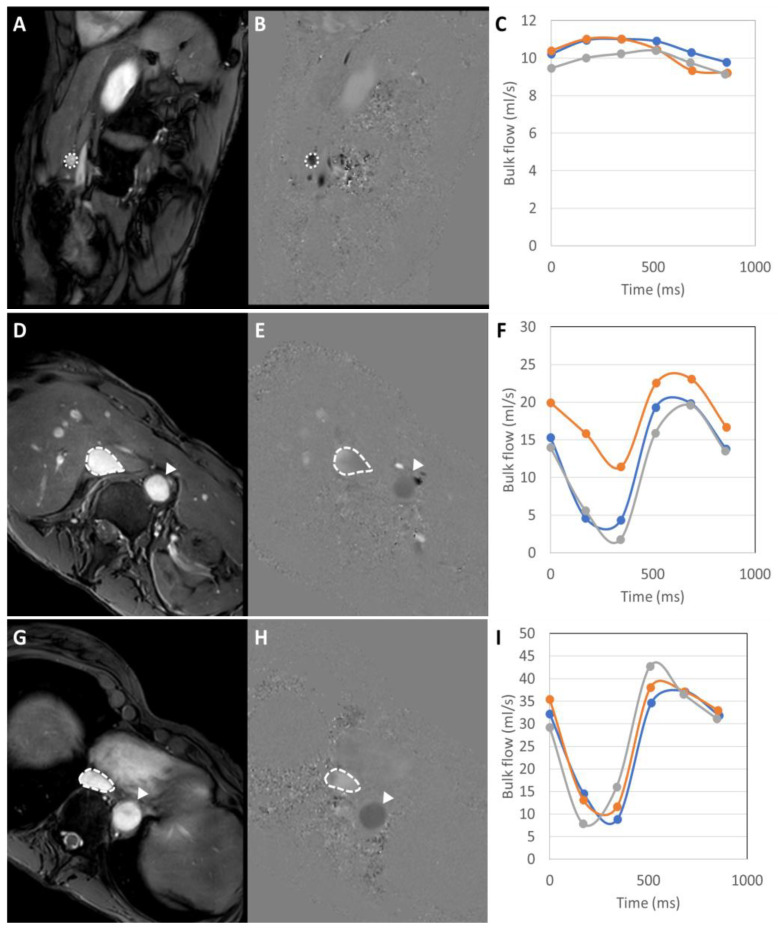
Two-dimensional phase-contrast measurements of bulk vessel flow in the portal vein (**A**–**C**), infrahepatic inferior vena cava (**D**–**F**) and suprahepatic inferior vena cava (**G**–**I**). Magnitude anatomical images (**A**,**D**,**G**) were used for vessel segmentation (dashed regions of interest), with segmentation of the portal vein (**A**), infrahepatic inferior vena cava (**D**) and suprahepatic inferior vena cava (**G**). These were transcribed onto matched velocity maps (**B**,**E**,**H**) and manually adjusted for seven-phases through the cardiac cycle. Measurements were performed in triplicate and corresponding bulk vessel flow across the cardiac cycle was averaged to estimate bulk flow in ml/min for the (**C**) portal vein (612 mL/min in this example), (**F**) infrahepatic inferior vena cava (858 mL/min in this example) and (**I**) suprahepatic inferior vena cava (1638 mL/min in this example). Total liver blood flow was estimated by subtracting infrahepatic from suprahepatic inferior vena cava flow (780 mL/min in this example). Bulk flow measurements were normalised to liver volume estimated from segmentation of the liver on anatomical imaging.

**Figure 6 nutrients-12-02151-f006:**
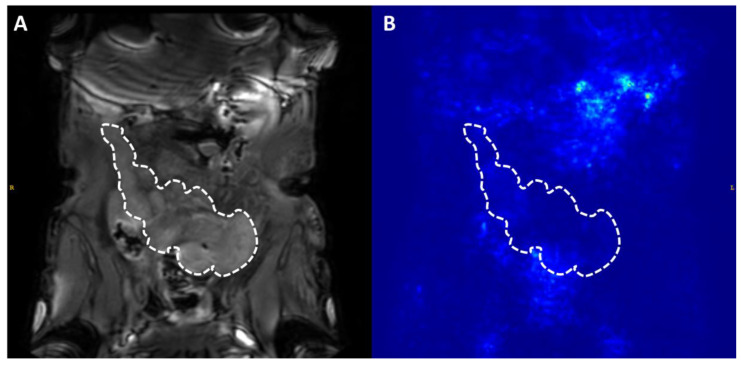
Dynamic cine MRI small bowel imaging was used to quantify small bowel motility. Regions of interest drawn on anatomical imaging (**A**) were colocalised to motility maps (the standard deviation of the determinant of pixel’s Jacobian) value (**B**). Averages were drawn across multiple slices for entire small bowel coverage. Small bowel motility in this example was 0.21 arbitrary units.

**Table 1 nutrients-12-02151-t001:** Selected combinations of biomarkers for the assessment of liver disease.

Name	Components	Liver Diseases in Which the Biomarkers Have Been Studied	Score Calculation
APRI	AST and Platelet Count	Chronic Hepatitis C, Chronic Hepatitis B, Non-alcoholic steatohepatitis (NASH), NAFLD, biliary atresia and IFALD	((AST level/AST upper level of normal)/platelet count)) × 100 − [AST upper level of normal = 40] [34]
AST/ALT Ratio	AST and ALT	Alcoholic liver disease, primary biliary cirrhosis, NAFLD and IFALD	AST/ALT
ELF	Hyaluronic Acid (HA), PIIINP and Tissue inhibitor of metalloproteinase 1 (TIMP-1)	Mixed chronic liver diseases, Chronic Hepatitis C, and primary biliary cirrhosis	(2.494 + 0.84 ln (C_HA_) + 0.735 ln(C_PIIINP_) + 0.391 ln(C_TIMP-1_) [81]
FIB-4	ALT, AST and Platelet Count	HIV/HCV coinfection, Chronic Hepatitis B, NAFLD and IFALD	(age × AST level/platelet count × √ALT) [48]
Forns Index	Age, GGT, Cholesterol and Platelet Count	Chronic Hepatitis C, Chronic Hepatitis B, and alcoholic liver disease.	(7.811 − 3.131 × ln(platelet count) + 0.781 × ln(GGT) + 3.467 × ln(age) − 0.014 × cholesterol) [51]
Fibrosis Index	Age, GGT, Cholesterol and Platelet Count	NAFLD	(−2.948 + 0.562 × Forns index + 0.288 × APRI + 0.006 × Platelet count (10^9^/L)) [49]
GPR	GGT and Platelet count	HIV/Chronic Hepatitis B, coinfection	((GGT level/GGT upper level of normal)/platelet count) × 100 GGT upper limit of normal: 40 (women) and 60 (men) [57]
NAFLD Fibrosis Score	Age, Hyperglycaemia, BMI, Platelet Count, Albumin, AST and ALT	NAFLD	(−1.675 + 0.037 × age (years) + 0.094 × BMI (kg/m^2^) + 1.13 × impaired fasting glucose/diabetes (yes = 1, no = 0) + 0.99 × AST/ALT ratio − 0.013 × platelet count − 0.66 × albumin) [58]

ALT = Alanine Transaminase; APRI = Aspartate transaminase-to-Platelet Ratio Index; AST = Aspartate Transaminase; BMI = Body Mass Index; ELF = Enhanced Liver Fibrosis; FIB-4 = Fibrosis 4; GGT = Gamma-Glutamyl Transferase; GPR = Gamma-glutamyl transferase-to-Platelet Ratio; HA = Hyaluronic Acid; HCV = Hepatitis C Virus; HIV = Human Immunodeficiency Virus; IFALD = Intestinal Failure-Associated Liver Disease; NAFLD = Non-alcoholic Fatty Liver Disease; NASH = Non-alcoholic Steatohepatitis; PIIINP = Procollagen III N-terminal Peptide; TIMP-1 = Tissue inhibitor of metalloproteinase 1.

**Table 2 nutrients-12-02151-t002:** Comparisons of clinical and nutritional characteristics between adult participants with IF diagnosed with IFALD or no IFALD by radiological examinations (mean ± SD and median (range)).

Parameters	Total (*n* = 20)	IFALD (*n* = 8)	Non-IFALD (*n* = 12)	*p*-Value
*Clinical characteristics*				
Age (years)	51.15 ± 17.30	58.13 ± 15.90	46.50 ± 17.23	0.145
Gender (Males:Females)	8:12	4:4	4:8	0.648
BMI (kg/m^2^)	21.27 ± 3.63	23.73 ± 3.36	19.91 ± 2.95	0.036
Oral Diet (Yes:No)	15:5	7:1	8:4	0.292
*Pathophysiological* *classification of IF*				
SBS-I—*n* (%)	2 (10%)	1 (8.3%)	1 (8.3%)	0.833
SBS-JC—*n* (%)	1 (5%)	1 (8.3%)	0	
SBS-JIC—*n* (%)	1 (5%)	0	1 (8.3%)	
Dysmotility—*n* (%)	8 (40%)	3 (37.5%)	5 (41.7%)	
Mechanical obstruction—*n* (%)	2 (10%)	0	2 (16.7%)	
Mucosal disease—*n* (%)	6 (30%)	3 (37.5%)	3 (50%)	
Small bowel length (cm)	100.56 ± 52.05	75.00 ± 49.12	132.50 ± 39.48	0.100
*Nutritional characteristics*				
PN duration (months)	88.90 (77.33)	85.50 (78.75)	57.50 (64.75)	0.980
Age PN started (years)	43.55 ± 17.61	47.50 ± 17.80	40.92 ± 17.76	0.428
PN mean energy (kcal/day)	1730.91 ± 372.02	1687.90 ± 419.00	1759.58 ± 353.67	0.685
PN mean lipids (g/kg/day)	0.41 ± 0.28	0.39 ± 0.30	0.41 ± 0.28	0.887
Days of PN/week	7 (2.50)	7 (0.25)	6.5 (3)	0.257
Days of PN lipids/week	2.80 ± 1.58	2.25 ± 1.16	3.17 ± 1.75	0.211

BMI = Body Mass Index; IF = Intestinal Failure; IFALD = Intestinal Failure-Associated Liver Disease; PN = Parenteral Nutrition; SBS-I = Short Bowel Syndrome with Ileostomy or Ileo-rectal anastomosis; SBS-JC = Short Bowel Syndrome with Jejunocolonic anastomosis; SBS-JIC = Short Bowel Syndrome with Jejunoileal anastomosis with an Intact Colon.

**Table 3 nutrients-12-02151-t003:** Comparisons of non-invasive biochemical parameters between adult participants with IF diagnosed with IFALD or no IFALD by radiological examinations (mean ± SD and median (range)).

Biochemical Parameters	Normal Range	Total (*n* = 20)	IFALD (*n* = 8)	Non-IFALD (*n* = 12)	*p*-Value
Platelet Count (× 10^9^/L)	150–400	219.80 ± 65.53	172.00 ± 42.04	251.67 ± 59.35	0.040
C-Reactive Protein (mg/L)	0–5.0	4.30 (8.00)	0.90 (1.23)	8.20 (9.70)	0.005
PIIINP (μg/L)	1.7–4.2	4.72 ± 2.26	6.20 ± 1.93	3.68 ± 1.92	0.018
Haptoglobin (g/L)	0.3–2.0	1.59 ± 0.96	0.88 ± 0.70	2.07 ± 0.81	0.003
Bilirubin(μmol/L)	0–20	7.35 (5.83)	11.84 (6.54)	6.07 (3.34)	0.005
AST (IU/L)	0–31	28.30 ± 11.41	38.88 ± 9.08	21.25 ± 6.06	<0.001
ALT (IU/L)	10–35	24.20 (19.48)	36.67 (18.72)	18.94 (10.82)	0.040
GGT (IU/L)	6–42	35.00 (84.46)	130.00 (93.25)	23.25 (25.75)	0.040

ALT = Alanine Transaminase; AST = Aspartate Transaminase; GGT = Gamma-Glutamyl Transferase; IFALD = Intestinal Failure-Associated Liver Disease; PIIINP = Procollagen III N-terminal Peptide.

**Table 4 nutrients-12-02151-t004:** Comparisons of non-invasive hepatic fibrosis scores between adult participants with IF diagnosed with IFALD or no IFALD by radiological examinations (mean ± SD and median (range)).

Hepatic Fibrosis Scores	Total (*n* = 20)	IFALD (*n* = 8)	Non-IFALD (*n* = 12)	*p*-Value
ELF	8.91 ± 1.34	9.71 ± 1.24	8.34 ± 1.14	0.032
APRI	0.30 (0.40)	0.63 (0.32)	0.19 (0.13)	<0.001
FIB-4	1.23 (0.60)	2.21 (1.63)	0.96 (0.71)	0.010
Forns Index	5.36 ± 2.55	7.48 ± 1.94	3.94 ± 1.83	0.001
GPR	0.35 (0.82)	1.20 (1.37)	0.20 (0.22)	0.002
Fibrosis Index	1.49 ± 1.21	2.46 ± 0.98	0.84 ± 0.87	0.001

APRI = Aspartate transaminase-to-Platelet Ratio Index; ELF = Enhanced Liver Fibrosis; FIB-4 = Fibrosis 4; GPR = Gamma-glutamyl transferase-to-Platelet Ratio; IFALD = Intestinal Failure-Associated Liver Disease.

**Table 5 nutrients-12-02151-t005:** Demographic characteristics of patients in the MRI Cohort.

Parameters	
*Clinical characteristics*	
Age (years)	52.3 ± 18.0
Gender (Males:Females)	2:8
BMI (kg/m^2^)	20.95 ± 4.38
*Pathophysiological classification of IF*	
SBS—*n* (%)	8 (80.0%)
Dysmotility—*n* (%)	2 (20.0%)
*Nutritional characteristics*	
PN duration (months)	120 ± 87
PN mean energy (kcal/day)	1151 ± 398
PN mean lipids (g/kg/day)	3.85 ±3.54

BMI = Body Mass Index; IF = Intestinal Failure; MRI = Magnetic Resonance Imaging; PN = Parenteral Nutrition; SBS = Short Bowel Syndrome.

**Table 6 nutrients-12-02151-t006:** Comparison of MRI imaging Scores against described normal values.

	Non-IFALD (*n* = 5)	IFALD Steatosis (*n* = 5)	*p*-Value
	Median (range)	Median (range)	
Liver Fat Fraction (%)	2.14 (1.4–4.2)	10.90 (2.2–27.4)	0.032
Liver Iron Concentration (μmol/g)	11.3 (10.4–38.9)	16.0 (11.3–46.7)	0.222
Liver T1 (ms)	715 (544–848)	740 (594–919)	0.873
Portal venous flow (qPV) (mL/min/100 g)	64.2 (45.6–137.6)	56.2 (37.5–93.0)	0.667
Estimated Total Liver Blood Flow (eTLBF) (mL/min/100 g)	91.8 (63.5–145.7)	62.1 (27.9–119.3)	0.151
Small Bowel Mean motility (a.u.)	0.19 (0.07–0.21)	0.16 (0.11–0.21)	0.999

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
