# Peer review of "Serum Scoring and Quantitative Magnetic Resonance Imaging in Intestinal Failure-Associated Liver Disease: A Feasibility Study"

_nutrients, 2020, doi:10.3390/nu12072151_

Round 1

Reviewer 1 Report

I read with interest this paper by Fragkos and colleagues on hepatic risk scores and the use of MRI in IFALD, which is a topic of considerable interest in this disease.

Overall, the manuscript is well-written, interesting and grounded in the state of the art, although the introduction could be shortened somewhat. The main limitation of this study is of course the limited number of patients included in this study, but as the authors mention themselves, the number of (identified) cases in any center is limited.

One point that needs some clarification is the level of fibrosis in the included patients and correlation with the serum risk scores. The authors state multiple times that none of the patients had significant liver fibrosis, which was reinforced by MRI investigation. However, all risk scores that were evaluated and elevated in these patients, such as the ELF, FIB-4, Fibrosis index… have been primarily developed for the detection of liver fibrosis (and not for detection of any underlying liver disease, inflammation or steatosis). If it holds true that these patients did not have significant liver fibrosis and portal hypertension, why were these scores aberrant? Platelet count for instance is often incorporated in these scores because it reflects portal hypertension secondary to liver fibrosis. The FIB-4 scores in almost all IFALD patients would be reflective of an intermediate or high risk of advanced liver fibrosis.
Despite the difficulty in including more patients, the addition of IFALD patients who do have liver fibrosis could clarify this issue. E.g. are the risk scores even higher in these patients?

Given the way in which IFALD is defined, is there a possibility that patients with pre-existing NAFLD have been included? Related to this point, do these patients with steatotic IFALD have characteristics of NAFLD, such as insulin resistance, metabolic syndrome…?

Author Response

Reviewer 1

Comment 1: I read with interest this paper by Fragkos and colleagues on hepatic risk scores and the use of MRI in IFALD, which is a topic of considerable interest in this disease.

Overall, the manuscript is well-written, interesting and grounded in the state of the art, although the introduction could be shortened somewhat.

Response: We thank the reviewer for this comment which we have now addressed. We have revised the Introduction section accordingly and shortened it. Changes appear in pages 2-4 in track changes.

Comment 2: The main limitation of this study is of course the limited number of patients included in this study, but as the authors mention themselves, the number of (identified) cases in any center is limited.

Response: We thank the reviewer for this comment. We agree with the reviewer that this is a limitation of the present study and have reported it as such in the limitations section of our study. Nevertheless, this may be balanced by the careful selection of the present sample size with good and similar pathophysiological and nutritional features.

Comment 3: One point that needs some clarification is the level of fibrosis in the included patients and correlation with the serum risk scores. The authors state multiple times that none of the patients had significant liver fibrosis, which was reinforced by MRI investigation. However, all risk scores that were evaluated and elevated in these patients, such as the ELF, FIB-4, Fibrosis index… have been primarily developed for the detection of liver fibrosis (and not for detection of any underlying liver disease, inflammation or steatosis). If it holds true that these patients did not have significant liver fibrosis and portal hypertension, why were these scores aberrant? Platelet count for instance is often incorporated in these scores because it reflects portal hypertension secondary to liver fibrosis. The FIB-4 scores in almost all IFALD patients would be reflective of an intermediate or high risk of advanced liver fibrosis.
Despite the difficulty in including more patients, the addition of IFALD patients who do have liver fibrosis could clarify this issue. E.g. are the risk scores even higher in these patients?

Response: We thank the reviewer for this comment which we have now addressed. We have added the following section to discuss this aspect of our results (page 13, lines 352-362):

“This appears initially counterintuitive given that these scores correlate with fibrosis to varying degrees in non-IFALD chronic liver diseases. However, the primary intention of previous association studies was with fibrosis or advanced liver disease, rather than with well-phenotyped steatosis cohorts. Additionally, it is possible that subclinical fibrosis was present in some patients in the present study; no patients had undergone prior liver biopsy. Furthermore, the pathophysiology of IFALD is much less well characterised than other chronic liver diseases. Whilst it is possible that the development of IFALD relates more closely to that of NAFLD than viral hepatitis, this has not been formally established. For example, little is known about vascular dynamics, the pathology of iron deposition, the function of hepatic stellate cells and other processes in IFALD. It is possible that some, or a combination, of these factors explain the serum score correlations found both in this small study and other non-IFALD studies.”

Comment 4: Given the way in which IFALD is defined, is there a possibility that patients with pre-existing NAFLD have been included? Related to this point, do these patients with steatotic IFALD have characteristics of NAFLD, such as insulin resistance, metabolic syndrome…?

Response: We thank the reviewer for this comment which we have now addressed. In our patient cohort, there was no evidence of liver disease (including NAFLD) prior to the development of intestinal failure in any patient. We have added the relevant sentence (page 7, line 242)

Reviewer 2 Report

Diverse liver damage often occurs in long-term parenteral treated patients. In adults is characterized by steatosis with progression to fibrosis. The aim of the study was to assess the clinical utility of complex serum scores and quantitative magnetic resonance imaging in patients with intestinal failure and with/without concomitant liver steatosis (IFALD). The number of patients included into the study was not very large, but carefully chosen, with good pathophysiological and nutritional characteristics. Compared to non-IFALD patients with INFALD-steatosis demonstrated statistically significant serum scores elevations, among others Aspartate transaminase – to- Platelet Ratio Index, Gamma-glutamyl transferase –to- Platelet Ratio Index, Fibrosis-4 Index, Forns Index. Moreover, the Quantitative MRI Scanning showed higher fat fraction in IFALD-steatosis patients. The obtained results contain clinical value and confirm that non-invasive methods can be useful for monitoring liver failure in patients with long-term home parenteral nutrition.

Conclusion: The article is well prepared in terms of content.

Author Response

Reviewer 2

Comments: Diverse liver damage often occurs in long-term parenteral treated patients. In adults is characterized by steatosis with progression to fibrosis. The aim of the study was to assess the clinical utility of complex serum scores and quantitative magnetic resonance imaging in patients with intestinal failure and with/without concomitant liver steatosis (IFALD). The number of patients included into the study was not very large, but carefully chosen, with good pathophysiological and nutritional characteristics. Compared to non-IFALD patients with INFALD-steatosis demonstrated statistically significant serum scores elevations, among others Aspartate transaminase – to- Platelet Ratio Index, Gamma-glutamyl transferase –to- Platelet Ratio Index, Fibrosis-4 Index, Forns Index. Moreover, the Quantitative MRI Scanning showed higher fat fraction in IFALD-steatosis patients. The obtained results contain clinical value and confirm that non-invasive methods can be useful for monitoring liver failure in patients with long-term home parenteral nutrition. Conclusion: The article is well prepared in terms of content.

Response: We thank the reviewer for this comment. No revision needed for this,

Round 2

Reviewer 1 Report

/